# Aldosterone synthase inhibitors in uncontrolled and resistant hypertension: A phenotype-stratified systematic review and network meta-analysis of randomized trials

Ismaila Ajayi Yusuf[1], Olurotimi J. Badero[2], Alozie Ihesiulo[3],
Micah Nnabuko Okwah[4]*, Abieyuwa Oshodin[5], Emmanuella Asikong[3]

1 Department of Internal Medicine, Guthrie Robert Packer Hospital, Sayre, Pennsylvania, United States of America, 2 Division of CardioNephrology, Cardiac Renal & Vascular Associates, Jackson, Mississippi, United States of America, 3 Department of Medicine, University of Calabar, Calabar, Nigeria, 4 Department of Epidemiology, Yale School of Public Health, New Haven, Connecticut, United States of America, 5 Department of Medicine, Obafemi Awolowo University, Ile-Ife, Nigeria

* micah.okwah@yale.edu

## Abstract

### Background

Aldosterone synthase inhibitors (ASIs) have emerged as a mechanistically targeted strategy for resistant and uncontrolled hypertension; however, no head-to-head trials exist, and comparative efficacy and safety remain uncertain. We compared their efficacy and safety using network and pairwise meta-analysis of randomized trials.

### Methods

This systematic review and meta-analysis adhered to the PRISMA guidelines. PubMed/MEDLINE, Scopus, Embase, ClinicalTrials.gov, and Cochrane Library were searched from inception to January 14, 2026, for randomized trials evaluating ASIs versus placebo or standard care. A frequentist random-effects network meta-analysis assessed systolic (SBP) and diastolic blood pressure (DBP). Dichotomous safety outcomes were pooled using Hartung–Knapp random-effects models. Network consistency was evaluated using design-by-treatment interaction modeling. Prespecified subgroup analyses stratified outcomes by hypertension phenotype (resistant vs uncontrolled).

### Results

Across 7 RCTs (n = 2,828), all ASIs significantly reduced SBP versus placebo: baxdrostat −8.63 mmHg (95% CI −10.84 to −6.42), lorundrostat −7.47 mmHg (95% CI −9.54 to −5.40), and LCI699/osilodrostat −5.63 mmHg (95% CI −9.15 to −2.12), with no significant indirect differences between agents. In resistant hypertension,

**Data availability statement:** All relevant data are within the paper and its Supporting Information files. Extracted data were derived from previously published randomized controlled trials, which are cited in the reference list.

**Funding:** The author(s) received no specific funding for this work.

**Competing interests:** The authors have declared that no competing interests exist.

lorundrostat (−9.00 mmHg; 95% CI −13.19 to −4.81) and baxdrostat (−8.77 mmHg; 95% CI −10.50 to −7.05) demonstrated pronounced reductions. In uncontrolled hypertension, LCI699/osilodrostat showed the largest point estimate (−10.55 mmHg; 95% CI −16.49 to −4.61), though this derives from a single early-phase trial and requires cautious interpretation. DBP reductions were significant for baxdrostat (−3.23 mmHg; 95% CI −4.73 to −1.73) and lorundrostat (−3.60 mmHg; 95% CI −5.43 to −1.77). Hypotension (RR 2.67), hyperkalemia (RR 7.94), and hyponatremia (RR 2.07) were significantly increased; serious adverse events, discontinuation, and network inconsistency were not detected.

## Conclusions

ASIs provide clinically meaningful BP reduction across both hypertension phenotypes; however, short-term use is associated with hypotension and electrolyte disturbances, necessitating careful monitoring. Phenotype-specific efficacy and long-term safety require validation in outcome-driven trials. Systematic Review Registration: PROSPERO CRD420251266257

## 1. Introduction

Hypertension remains the leading modifiable determinant of global cardiovascular morbidity and mortality, affecting approximately 1.4 billion individuals worldwide [1]. Despite advances in antihypertensive pharmacotherapy and public health interventions, a substantial proportion of treated patients fail to achieve adequate blood pressure control. Uncontrolled hypertension, defined as blood pressure (BP) ≥130/80 mmHg despite antihypertensive therapy, affects up to 55% of treated individuals [1]. True resistant hypertension (RHT), characterized by persistent elevation despite maximally tolerated therapy with three mechanistically distinct agents, typically including a renin–angiotensin system blocker, a long-acting calcium channel blocker, and a diuretic, or achievement of control only with four or more agents, occurs in up to 10% of treated patients after exclusion of non-adherence, therapeutic inertia, and white-coat effect [1–5]. Both uncontrolled and resistant phenotypes confer markedly elevated risks of heart failure, ischemic heart disease, stroke, and progressive renal dysfunction [6], underscoring a critical unmet therapeutic need. Dysregulation of the renin–angiotensin–aldosterone system (RAAS) represents a central pathophysiologic driver of persistent hypertension [7,8].

Excess aldosterone promotes sodium retention, vascular remodeling, myocardial fibrosis, inflammation, and endothelial dysfunction, contributing not only to sustained blood pressure elevation but also to cardiovascular and renal end-organ injury [9–11]. In resistant and uncontrolled hypertension, RAAS dysregulation may account for persistent elevation in approximately 30% of affected patients despite multidrug therapy [9,10]. Mineralocorticoid receptor antagonists (MRAs), such as spironolactone and eplerenone, are recommended as fourth-line therapy in RHT [10,12]; however, their

clinical utility is constrained by an increased risk of hyperkalemia, potential deterioration in renal function, and hormone-related adverse effects, which may limit tolerability and broader implementation.

Moreover, MRAs do not suppress aldosterone synthesis and may permit compensatory increases in renin and circulating aldosterone, potentially sustaining receptor-independent signaling pathways [12]. Although non-steroidal MRAs mitigate some endocrine adverse effects, important therapeutic gaps remain [10].

Selective inhibition of aldosterone synthase (CYP11B2), the enzyme responsible for the terminal steps of aldosterone biosynthesis in the adrenal cortex, has therefore emerged as a mechanistically targeted strategy [11–13]. Aldosterone synthase inhibitors (ASIs), including baxdrostat, lorundrostat, and osilodrostat/LCI699, directly reduce circulating aldosterone levels, offering more complete suppression of aldosterone signaling while minimizing off-target cortisol inhibition observed with earlier non-selective agents [13–17]. Early-phase randomized trials provide proof of concept for this approach. In the phase 2 BrigHTN trial, baxdrostat produced dose-dependent reductions in systolic blood pressure, with the highest dose achieving an approximate 20 mmHg greater reduction compared with placebo (95% CI −16.4 to −5.5; P < 0.001) [15]. Similarly, phase 2 studies of lorundrostat and osilodrostat/LCI699 demonstrated significant reductions in office and ambulatory blood pressure in patients with uncontrolled or resistant hypertension receiving background therapy [9]. Collectively, these findings support aldosterone synthase inhibition as a promising therapeutic strategy for persistent hypertension.

However, no head-to-head randomized controlled trials have directly compared available aldosterone synthase inhibitors (ASIs). Existing systematic reviews and meta-analyses have largely synthesized early-phase trials, frequently pooled resistant and uncontrolled hypertension phenotypes, modeled multiple dose regimens within heterogeneous populations, and often reported safety outcomes descriptively or predated contemporary phase 3 evidence [18–23]. Consequently, the comparative efficacy, safety profiles, and optimal therapeutic positioning of individual ASIs within contemporary hypertension management remain uncertain. Accordingly, we conducted an updated network meta-analysis of randomized controlled trials evaluating baxdrostat, lorundrostat, and osilodrostat/LCI699 in patients with resistant and uncontrolled hypertension. By integrating early- and late-phase randomized evidence, applying a phenotype-stratified comparative framework (resistant vs uncontrolled hypertension), and triangulating inference through parallel network and pairwise analyses with phase-restricted sensitivity testing, we aimed to compare blood pressure–lowering efficacy, treatment ranking probabilities, and safety outcomes using quantitative meta-analysis of clinically actionable adverse events in contemporary cardiovascular practice.

## 2. Methods

### 2.1. Study design and reporting standards

This systematic review and network meta-analysis of randomized controlled trials was conducted in accordance with the Preferred Reporting Items for Systematic Reviews (Fig 1) and Meta-Analyses (PRISMA 2020) statement and the PRISMA extension for network meta-analysis [24,25]. The study protocol was prospectively registered with PROSPERO (CRD420251266257) before data extraction and is available in the supplementary materials (S1 File).

Ethical approval and informed consent were not required for this study because it is a systematic review of previously published studies.

### 2.2. Data sources and search strategy

A comprehensive literature search was performed in MEDLINE, Embase, the Cochrane Central Register of Controlled Trials (CENTRAL), Scopus, and ClinicalTrials.gov from database inception through January 14, 2026. The search strategy combined controlled vocabulary and free-text terms related to aldosterone synthase inhibition and hypertension, including "aldosterone synthase inhibitor," "baxdrostat," "lorundrostat," "osilodrostat," "LCI699," "resistant hypertension," and "uncontrolled hypertension."

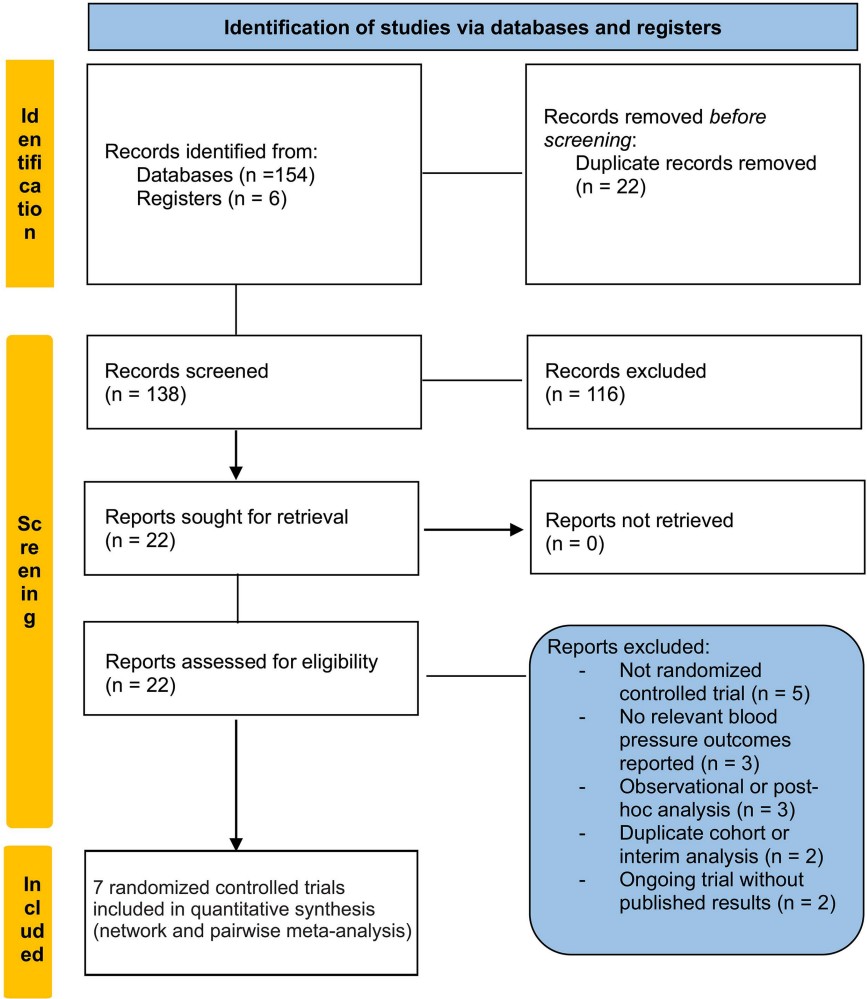

**Fig 1. PRISMA Flow Diagram.**

The PICOS (Population, Intervention, Comparator, Outcomes, Study design) framework guided development of the search strategy. Reference lists of relevant articles and prior reviews were manually screened to identify additional eligible studies. ClinicalTrials.gov was queried to identify unpublished or ongoing trials. The complete search strategy is provided in the Supplement (S2 File).

## 2.3. Study selection

Eligible studies met the following criteria:

1. Randomized controlled trial design

2. Adult participants (≥18 years) with resistant or uncontrolled hypertension

3. Comparison of an aldosterone synthase inhibitor (baxdrostat, lorundrostat, osilodrostat/LCI699) versus standard of care

4. Reported quantitative blood pressure outcomes

Resistant hypertension was defined according to guideline-based criteria as persistent blood pressure elevation despite maximally tolerated therapy with three antihypertensive agents of complementary mechanisms (including a diuretic), or control achieved with four or more agents. Uncontrolled hypertension was defined as blood pressure ≥130/80 mmHg despite ongoing pharmacologic therapy.

Two independent reviewers (I.A.Y and A.I) screened titles and abstracts, followed by a full-text review of potentially eligible studies. Disagreements were resolved by consensus or adjudication by a third reviewer (M.N.O). Reasons for exclusion at the full-text stage were documented and are presented in the PRISMA flow diagram.

## 2.4. Data extraction

Data were extracted independently by two reviewers (I.A.Y and A.I) using a standardized data collection form, with discrepancies resolved by a third reviewer (M.N.O). Extracted variables included:

- Study design and phase

- Sample size

- Participant demographics and baseline blood pressure

- Intervention dose and duration

- Comparator characteristics

- Mean changes in systolic and diastolic blood pressure

- Event counts for safety outcomes

When necessary, corresponding authors were contacted for clarification. For multi-arm trials, shared comparator groups were appropriately accounted for to prevent double-counting.

## 2.5. Study outcomes

### 2.5.1. Primary outcome.

- Change in systolic blood pressure (mean difference, mmHg).

### 2.5.2. Secondary outcomes.

- Change in diastolic blood pressure

- Hypotension

- Hyperkalemia

- Hyponatremia

- Serious adverse events

- Discontinuation due to adverse events

Adverse events were extracted as reported in primary publications. Given heterogeneity in safety definitions across trials, including variation in laboratory thresholds for hyperkalemia and hyponatremia, event counts were extracted using the closest available approximation to a common threshold, with the most conservative threshold applied where definitions differed materially.

Subgroup analyses were prespecified for resistant and uncontrolled hypertension phenotypes.

## 2.6. Quality assessment

### 2.6.1. Risk of bias assessment.
Risk of bias was evaluated using the Cochrane Risk of Bias 2.0 tool across five domains (Fig 2):

1. Randomization process

2. Deviations from intended interventions

3. Missing outcome data

4. Measurement of outcomes

5. Selection of reported results

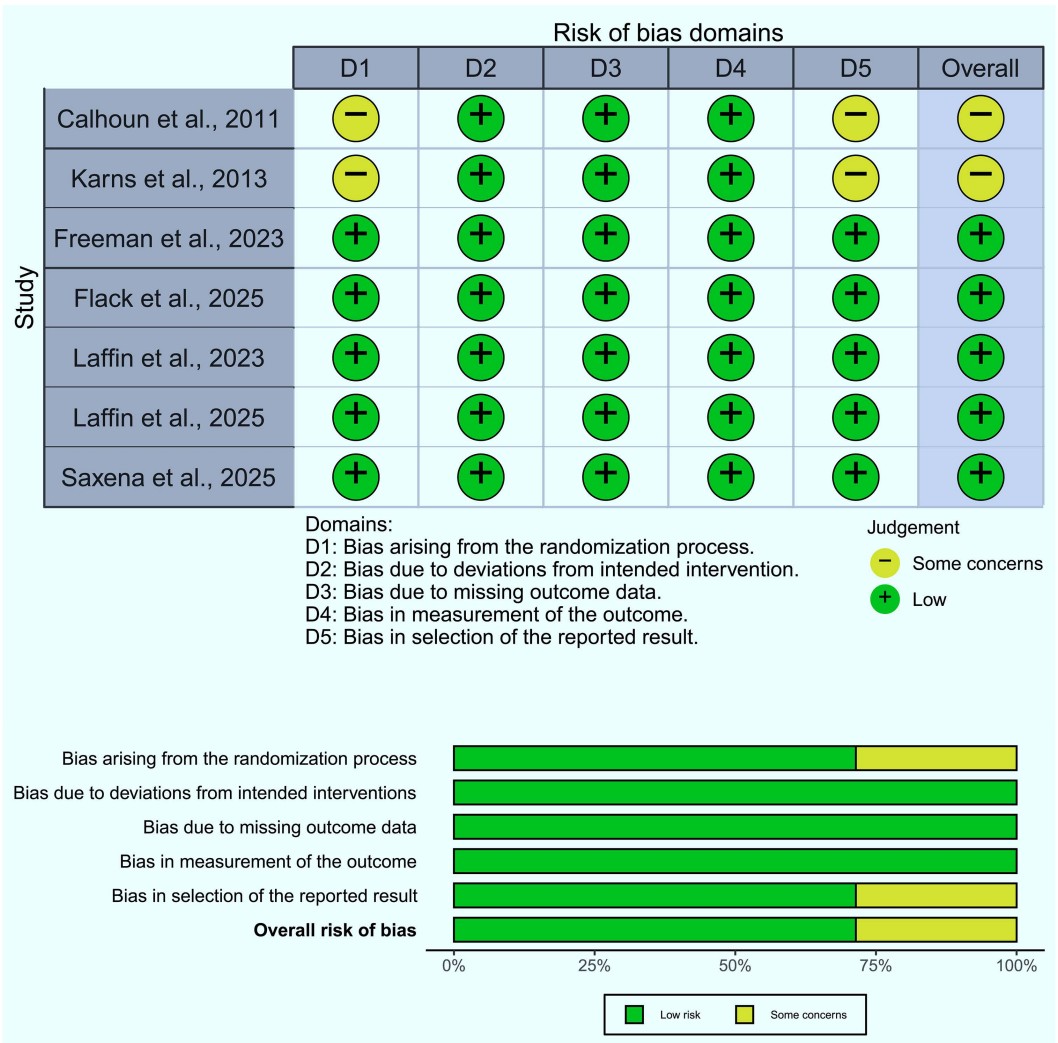

**Fig 2. Risk of bias summary.**

Each study was categorized as low risk, some concerns, or high risk of bias. Disagreements were resolved by consensus.

**2.6.2. GRADE certainty of evidence.** The certainty of evidence for each outcome was assessed using the Grading of Recommendations Assessment, Development and Evaluation (GRADE) approach. Evidence from randomized controlled trials was initially rated as high certainty and downgraded based on risk of bias, inconsistency, indirectness, imprecision, and publication bias. GRADE ratings were independently assessed by two reviewers (I.A.Y and A.I), with discrepancies resolved by a third reviewer (M.N.O). The overall certainty of evidence for each outcome was categorized as high, moderate, low, or very low. Findings are reported in the results section of this study. For network estimates, GRADE certainty ratings were derived by integrating direct and indirect evidence, with downgrading guided by the higher risk of bias and imprecision among contributing comparisons.

### 2.7. Statistical analysis

**2.7.1. Pairwise meta-analysis.** For direct comparisons, random-effects meta-analyses were performed using the Hartung–Knapp–Sidik–Jonkman method to provide more conservative variance estimates. Continuous outcomes were pooled as mean differences (MD) with 95% confidence intervals (CI). Dichotomous outcomes were pooled as risk ratios (RR). Between-study heterogeneity was quantified using $\tau^2$ and $I^2$ statistics. Cochran's Q test assessed statistical heterogeneity.

**2.7.2. Network meta-analysis.** A frequentist random-effects network meta-analysis was conducted to compare all interventions simultaneously while preserving randomization within trials. The transitivity assumption was evaluated by assessing the similarity of baseline characteristics and study design across treatment comparisons.

Consistency between direct and indirect evidence was evaluated using the design-by-treatment interaction model. Global inconsistency was assessed via between-design Q statistics.

Relative treatment ranking was estimated using P-scores, representing the mean extent of certainty that a treatment is superior to competing interventions.

Subgroup and Sensitivity Analyses.

Prespecified subgroup analyses were performed by:

• Individual drug (baxdrostat, lorundrostat, osilodrostat/LCI699)

• Hypertension phenotype (resistant vs uncontrolled)

**2.7.3. Sensitivity analyses.** Sensitivity analyses were conducted to ensure robustness of outcomes.

1. Leave-one-out analysis, sequentially omitting each study

2. Comparison of common-effect versus random-effects models

Subgroup differences were interpreted cautiously and considered meaningful only when supported by consistent directionality and statistical interaction testing. Funnel plots were visually inspected when 10 or more studies were available. Due to the limited number of trials per comparison, formal Egger regression testing was not consistently performed. All analyses were performed using R (version 4.5.2) with the "meta" and "netmeta" packages. A two-sided *p*-value <0.05 was considered statistically significant.

**2.7.4. Assessment of transitivity.** The validity of network meta-analysis relies on the transitivity assumption, that trials comparing different treatments are sufficiently similar to allow indirect comparisons. We evaluated transitivity by comparing baseline characteristics of included trials across treatment comparisons. Domains assessed included study design and phase, hypertension phenotype and population, sample size, treatment arms and doses, trial duration, baseline blood pressure and age, and background antihypertensive therapy. Distributions were compared descriptively

across treatment comparisons, and substantial imbalances that could threaten the transitivity assumption were identified and evaluated. Potential effect modifiers were selected a priori based on biological plausibility and prior evidence from hypertension trials, recognizing that aldosterone-driven pathophysiology, background therapy intensity, and hypertension phenotype are established determinants of antihypertensive response.

**2.7.5. Handling of multiple dose arms.** When trials evaluated multiple dose arms of the same aldosterone synthase inhibitor, and no single dose was prespecified as the primary clinical target, active dose groups were combined into a single pooled intervention arm. Continuous outcomes were pooled using Cochrane-recommended methods to calculate a weighted mean and pooled standard deviation, while dichotomous outcomes were combined by summing events and total participants across dose arms [26]. This approach preserved within-trial randomization, maintained the full sample size, and avoided artificial inflation of precision arising from multiple comparisons against a shared control group. As a consequence of dose pooling, formal dose-response modeling was not performed within the meta-analytic framework. Dose-specific efficacy and safety data from the primary publications did not reveal systematic differences that would contraindicate pooling; where dose-dependent patterns were identified, these are summarized in the results section of this study (section 3.7), based on reported data from the primary trial publications.

**2.7.6. Study cohort and analytical structure.** Randomized controlled trials evaluating selective aldosterone synthase inhibition with baxdrostat, lorundrostat, and LCI699 (osilodrostat) in resistant and uncontrolled hypertension were included in pairwise and network meta-analyses. Continuous outcomes were expressed as mean differences (MD) in systolic and diastolic blood pressure (SBP and DBP). Dichotomous outcomes were pooled using Hartung–Knapp random-effects models and reported as risk ratios (RRs). Network coherence was assessed via design-by-treatment interaction modeling, and heterogeneity was quantified using $\tau^2$ and $I^2$ statistics. Prespecified leave-one-out sensitivity analyses were conducted to evaluate robustness.

## 3. Results

### 3.1. General characteristics of included studies

Baseline characteristics of the seven included trials (n = 2828) are presented in Table 1. Across studies, mean baseline SBP ranged from approximately 145–152 mmHg, and mean age clustered between 55 and 62 years, indicating broadly comparable hypertension severity and demographic distribution.

Although the proportion of participants with resistant hypertension varied—BrigHTN enrolling exclusively resistant hypertension, whereas phase 3 programs such as Launch-HTN and BaxHTN included mixed uncontrolled and resistant populations—the distribution of baseline systolic blood pressure and antihypertensive burden remained overlapping across studies. Because the resistant phenotype may act as an effect modifier, prespecified phenotype-stratified analyses were performed. Overall, the similarity in key clinical and methodological characteristics supports the plausibility of the transitivity assumption while acknowledging modest heterogeneity in phenotype composition that was addressed analytically.

### 3.2. GRADE certainty of evidence

Certainty of evidence was evaluated using GRADE (Table 2). For overall systolic blood pressure (SBP) reduction, the evidence was of high certainty, with no serious concerns across risk of bias, inconsistency ($I^2$ = 41.5%; no incoherence), indirectness, imprecision (CIs excluded the null value), or publication bias. Subgroup analyses in resistant and uncontrolled hypertension likewise demonstrated high-certainty evidence. Diastolic blood pressure (DBP) reduction was downgraded to moderate certainty due to serious inconsistency ($I^2$ = 92.5%). Hyperkalemia was rated moderate certainty, downgraded for imprecision due to the wide confidence interval reflecting limited event counts across trials, despite a significant pooled estimate. Hyponatremia was similarly rated moderate certainty, downgraded for imprecision, given substantial

**Table 1. General characteristics of included randomized trials.**

| s | Drug | Phase | Population | N randomized | Treatment arms | Duration | Baseline BP/ age | Background therapy |
|---|---|---|---|---|---|---|---|---|
| Study 1 | LCI699/ osilodrostat | 2 | Uncontrolled Primary HTN | 524 | LCI699 0.25 mg QD, 0.5 mg QD, 1 mg QD, 0.5 mg BID; eplerenone 50 mg BID; placebo | 8 wk | Mean BP 157.9/100.2 mmHg; mean age 55 yr | Untreated or ≤2 agents |
| Study 2 | LCI699/ osilodrostat | 2 | Resistant HTN | 155 | LCI699 0.25 mg BID, 1 mg QD, 0.5→1 mg BID; eplerenone 50 mg BID; placebo | 8 wk | Mean BP 152.9/89.8 mmHg; mean age 56.5 yr | Stable ≥3-drug regimen including diuretic |
| Study 3 | Baxdrostat | 2 | Treatment-resistant HTN | 275 | Baxdrostat 0.5, 1, or 2 mg QD; placebo | 12 wk | Mean baseline office BP NR; eligibility BP ≥ 130/80 mmHg | ≥3 agents including diuretic |
| Study 4 | Lorundrostat | 2 | Uncontrolled HTN | 200 | Lorundrostat 12.5, 50, 100 mg QD; 12.5 or 25 mg BID; placebo | 8 wk | Mean baseline BP NR; eligibility AOBP SBP ≥ 130 mmHg | ≥2 antihypertensive agents |
| Study 5 | Lorundrostat | 2b | Confirmed uncontrolled HTN | 285 | Lorundrostat 50 mg QD; 50→100 mg QD; placebo | 12 wk | Office SBP ~ 155 mmHg; 24-h SBP ~ 140 mmHg; mean age 60 yr | Standardized 2- or 3-drug regimen |
| Study 6 | Lorundrostat | 3 | Uncontrolled ± treatment-resistant HTN | 1083 | Lorundrostat 50 mg QD for 12 wk; 50→100 mg QD at week 6 if SBP ≥ 130; placebo | 12 wk | Mean Age 61.6 yr; 60.1% on ≥3 meds | 2–5 prescribed agents |
| Study 7 | Baxdrostat | 3 | Uncontrolled or resistant HTN | 796 | Baxdrostat 1 mg QD, 2 mg QD; placebo | 12 wk | Mean BP 149/87 mmHg; mean age 61 yr | 2 agents for uncontrolled HTN or ≥3 for resistant HTN, including diuretic |

**Footnote:** 3,318 randomized across 7 trials; 2,828 contributed extractable outcome data to pooled analyses.

Study identifiers are anonymized for compliance with journal data sharing policies; corresponding trials are fully detailed in the reference list.

**Abbreviations:** AOBP, automated office blood pressure; BID, twice daily; BP, blood pressure; HTN, hypertension; NR, not reported in extracted table fields; QD, once daily; SBP, systolic blood pressure.

dependence on a single trial. Serious adverse events and discontinuation due to adverse events were rated moderate certainty due to imprecision, with confidence intervals either crossing or approaching 1 in sensitivity analyses. Overall, efficacy outcomes were of high certainty, while safety outcomes were predominantly of moderate certainty.

### 3.3. Comparative effects on systolic blood pressure

**3.3.1. Overall network analysis.** Across the entire hypertensive cohort, all aldosterone synthase inhibitors produced statistically significant and clinically meaningful reductions in SBP compared with placebo (Fig 3). Baxdrostat reduced SBP by −8.63 mmHg (95% CI −10.84 to −6.42), lorundrostat by −7.47 mmHg (95% CI −9.54 to −5.40), and LCI699 (osilodrostat) by −5.63 mmHg (95% CI −9.15 to −2.12). The magnitude of reduction with baxdrostat and lorundrostat approached or exceeded the 7–10 mmHg threshold historically associated with substantial cardiovascular risk reduction in resistant populations [27–33].

Indirect comparisons between active agents did not demonstrate statistically significant differences. The mean difference between baxdrostat and lorundrostat was −1.16 mmHg (95% CI −4.19 to 1.87), and between baxdrostat and LCI699 (osilodrostat) was −3.00 mmHg (95% CI −7.15 to 1.15). Lorundrostat versus LCI699 (osilodrostat) yielded 1.84 mmHg (95% CI −2.24 to 5.91). All indirect contrasts crossed zero, indicating no conclusive superiority among active agents within available precision.

**Table 2. GRADE certainty of evidence.**

| Outcome | Risk of Bias | Inconsistency | Indirectness | Imprecision | Publication Bias | Overall Certainty |
|---|---|---|---|---|---|---|
| Systolic BP (Overall Network) | Not serious | Not serious (I²=41.5%; no incoherence) | Not serious | Not serious (CIs exclude null) | Undetected | **High** |
| Systolic BP – Resistant HTN | Not serious | Not serious | Not serious | Not serious | Undetected | **High** |
| Systolic BP – Uncontrolled HTN | Not serious | Not serious | Not serious | Not serious | Undetected | **High** |
| Diastolic BP (Overall Network) | Not serious | **Serious** (I²=92.5%) | Not serious | Not serious | Undetected | **Moderate** |
| Hypotension | Not serious | Not serious (I²=0%) | Not serious | Not serious (CI excludes 1) | Undetected | **High** |
| Hyperkalemia | Not serious | Not serious (I²=0%) | Not serious | **Serious** (wide CI; limited event counts) | Undetected | **Moderate** |
| Hyponatremia | Not serious | Not serious (I²=0%) | Not serious | **Serious** (single-trial dependence) | Undetected | **Moderate** |
| Serious Adverse Events | Not serious | Not serious (I²=0%) | Not serious | **Serious** (CI crosses 1) | Undetected | **Moderate** |
| Discontinuation Due to AEs | Not serious | Not serious (I²=0%) | Not serious | **Serious** (fragile; single-trial dependent) | Undetected | **Moderate** |

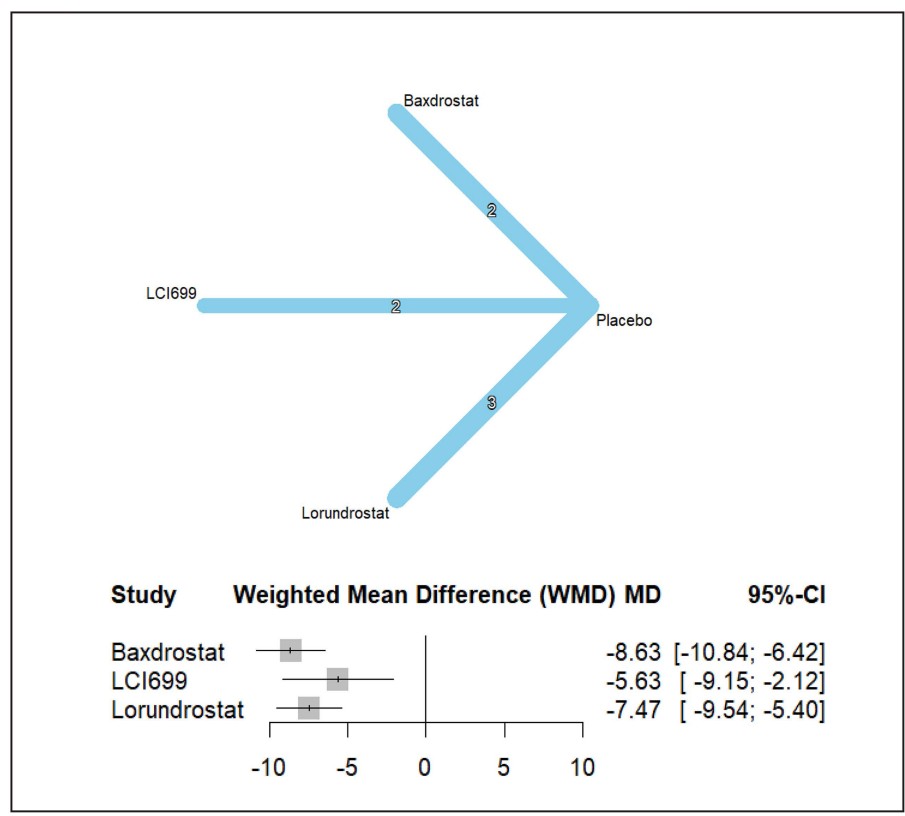

**Fig 3. Systolic Blood Pressure network and forest plots..**

Between-study heterogeneity in the SBP network was moderate (τ²=1.5434; τ=1.2423; I²=41.5%), with no statistically significant residual heterogeneity within designs (Q=6.83, p=0.145). Importantly, no inconsistency was observed between direct and indirect evidence (between-design Q=0.00), supporting the internal validity of the network model and the satisfaction of transitivity assumptions.

Ranking probability analysis under random-effects modeling identified lorundrostat as having the highest likelihood of being the most effective intervention (P-score 0.8985), followed by baxdrostat (0.6792), LCI699 (osilodrostat) (0.422), and placebo (0.0003). The ranking pattern remained stable under common-effects modeling, reinforcing the robustness of comparative efficacy ordering (S3 File).

### 3.4. Diastolic blood pressure effects

In the DBP network, both baxdrostat and lorundrostat demonstrated statistically significant reductions relative to placebo (Fig 4). LCI699/osilodrostat trials did not report extractable diastolic blood pressure outcomes and were therefore excluded from the DBP network analysis. Baxdrostat reduced DBP by −3.23 mmHg (95% CI −4.73 to −1.73), while lorundrostat reduced DBP by −3.60 mmHg (95% CI −5.43 to −1.77). These magnitudes are directionally consistent with the observed systolic effects and reflect a coherent hemodynamic profile across agents.

Direct comparison between baxdrostat and lorundrostat showed no statistically significant difference (MD 0.37 mmHg; 95% CI −1.99 to 2.74). Heterogeneity in the DBP network was substantial (τ²=1.1272; I²=92.5%), indicating considerable variability in diastolic response across trials. However, inconsistency between direct and indirect estimates was

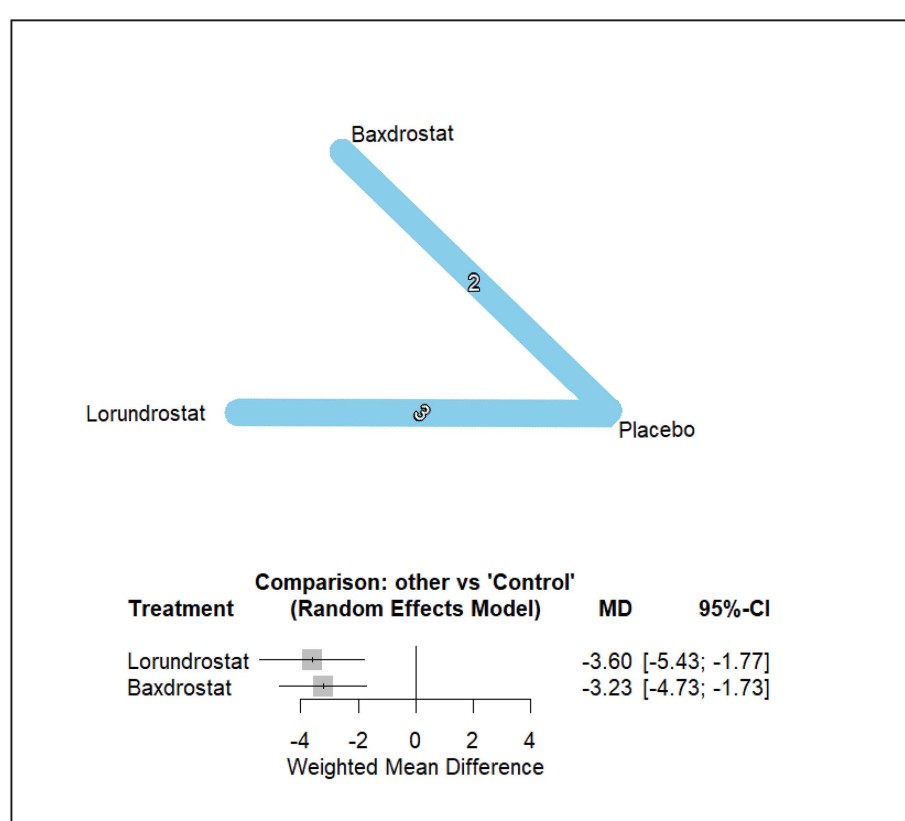

**Fig 4. Diastolic Blood Pressure network and forest plots.**

not detected (between-design Q = 0.00), suggesting that variability arose from between-study differences rather than structural incoherence within the network. Under random-effects ranking, lorundrostat had a P-score of 0.8108 and baxdrostat 0.6891, whereas the placebo ranked lowest. Despite high heterogeneity, the relative ordering of active treatments remained consistent. A post hoc exploratory meta-regression examining trial phase, background antihypertensive agent count, mean baseline SBP, and study duration did not identify any covariate that significantly explained the observed heterogeneity; residual variability likely reflects between-trial differences in background therapy intensity and diastolic physiology.

### 3.5. Subgroup analysis: resistant hypertension

**3.5.1. Continuous systolic reduction.** In the resistant hypertension subgroup, all three aldosterone synthase inhibitors demonstrated statistically significant SBP reductions compared with placebo (Fig 5). Lorundrostat reduced SBP by −9.00 mmHg (95% CI −13.19 to −4.81), and baxdrostat by −8.77 mmHg (95% CI −10.50 to −7.05). These effect sizes are notable given the refractory physiology characteristic of resistant disease. LCI699 (osilodrostat) also produced a statistically significant reduction (−3.53 mmHg; 95% CI −6.95 to −0.11), though of comparatively smaller magnitude. The overall pooled SBP reduction across all agents in resistant hypertension was −7.46 mmHg (95% CI −10.02 to −4.91). Indirect comparison between baxdrostat and lorundrostat showed no significant difference (MD 0.23 mmHg; 95% CI −4.59 to 5.04). Ranking probabilities were closely aligned between lorundrostat (P-score 0.8342) and baxdrostat (0.8189), both markedly higher than LCI699 (osilodrostat) (0.3357), consistent with the smaller absolute reduction observed with the latter agent. Overall heterogeneity across agents was moderate (I² = 71.5%), with significant subgroup differences detected between drugs (χ² = 7.52, df = 2, $p$ = 0.0233), suggesting that effect magnitude differs meaningfully across agents within this phenotype.

### 3.6. Subgroup analysis: uncontrolled hypertension

In uncontrolled hypertension, baxdrostat trials did not contribute separable subgroup data and were therefore excluded from this analysis. Among eligible trials, LCI699 (osilodrostat) demonstrated the largest point estimate for SBP reduction (−10.55 mmHg; 95% CI −16.49 to −4.61), exceeding that observed with lorundrostat (−7.04 mmHg; 95% CI −8.08 to −5.99), as seen in (Fig 6). However, the LCI699 estimate derives entirely from a single early-phase trial (Calhoun 2011) carrying 5.3% of the total weight with a standard error of 3.03, and should therefore be interpreted cautiously. The overall pooled SBP reduction across both agents in uncontrolled hypertension was −7.38 mmHg (95% CI −8.78 to −5.99). The indirect comparison between LCI699 (osilodrostat) and lorundrostat was not statistically significant (MD −3.51 mmHg; 95% CI −9.54 to 2.52), and the formal test for subgroup differences did not reach significance (χ² = 1.30, df = 1, $p$ = 0.2536), indicating that the observed difference in point estimates between agents is not statistically distinguishable. Ranking analysis identified LCI699 (osilodrostat) as having the highest probability of being most effective in this subgroup (P-score 0.9365), followed by lorundrostat (0.5634); however, given the single-trial basis of the LCI699 estimate and the non-significant subgroup difference test, this ranking should be considered hypothesis-generating rather than definitive. These findings suggest potential phenotype-specific differences in responsiveness between resistant and uncontrolled hypertension that warrant prospective validation in adequately powered trials.

### 3.7. Dose-response patterns across aldosterone synthase inhibitors

Because doses were pooled in the primary meta-analysis per Cochrane guidance, formal dose-response modeling was not performed. Within-trial dose-specific data from the primary publications are summarized descriptively in Table 3. A consistent pattern emerged across agents: higher doses generally produced greater placebo-adjusted SBP reductions, with attenuation at upper dose ranges. Baxdrostat demonstrated a graded response across 0.5–2 mg once daily, with a near-plateau effect at 2 mg in both BrigHTN and BaxHTN. Lorundrostat showed optimal placebo-adjusted reductions at

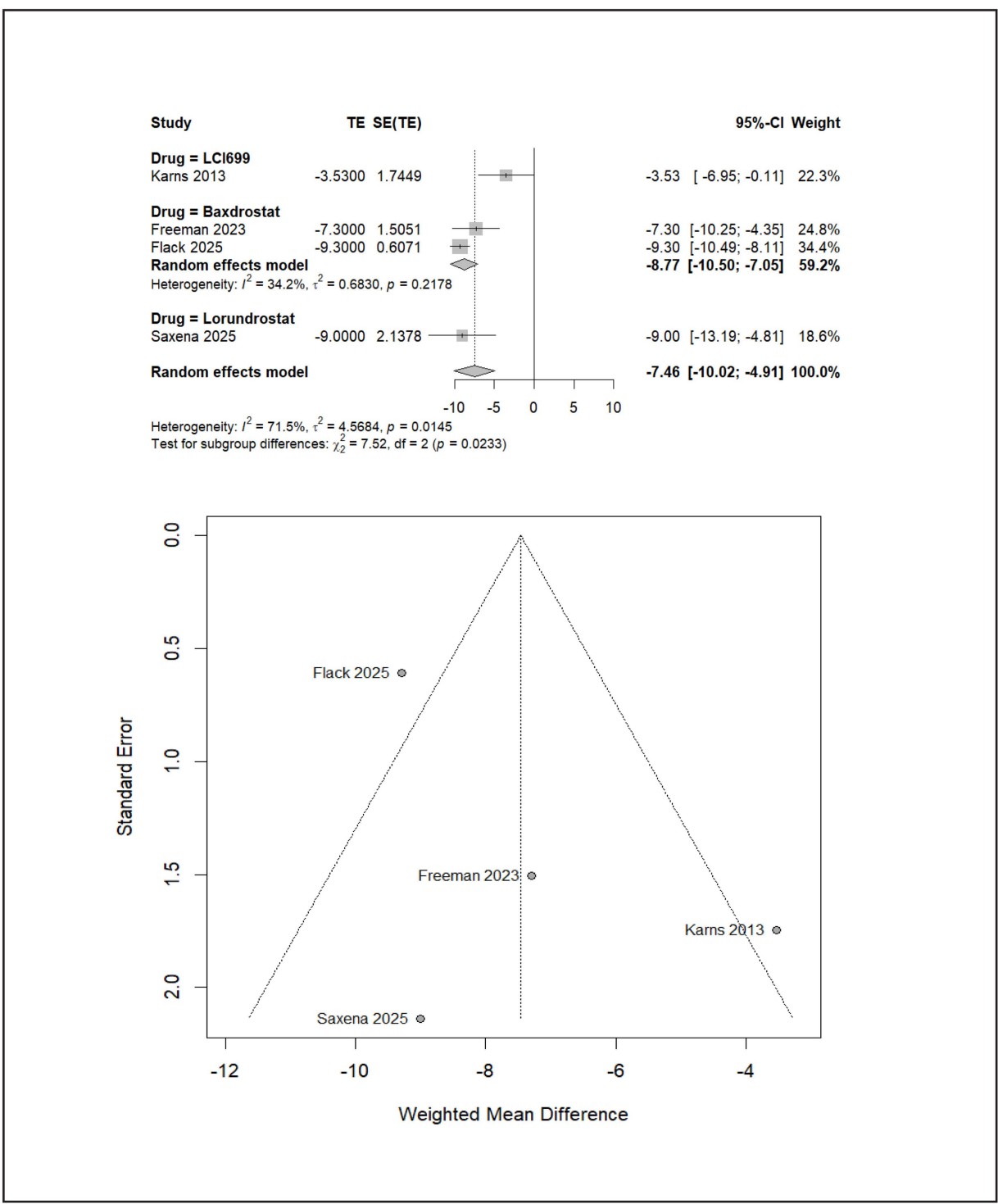

**Fig 5. Resistant hypertension funnel and forest plot.**

---

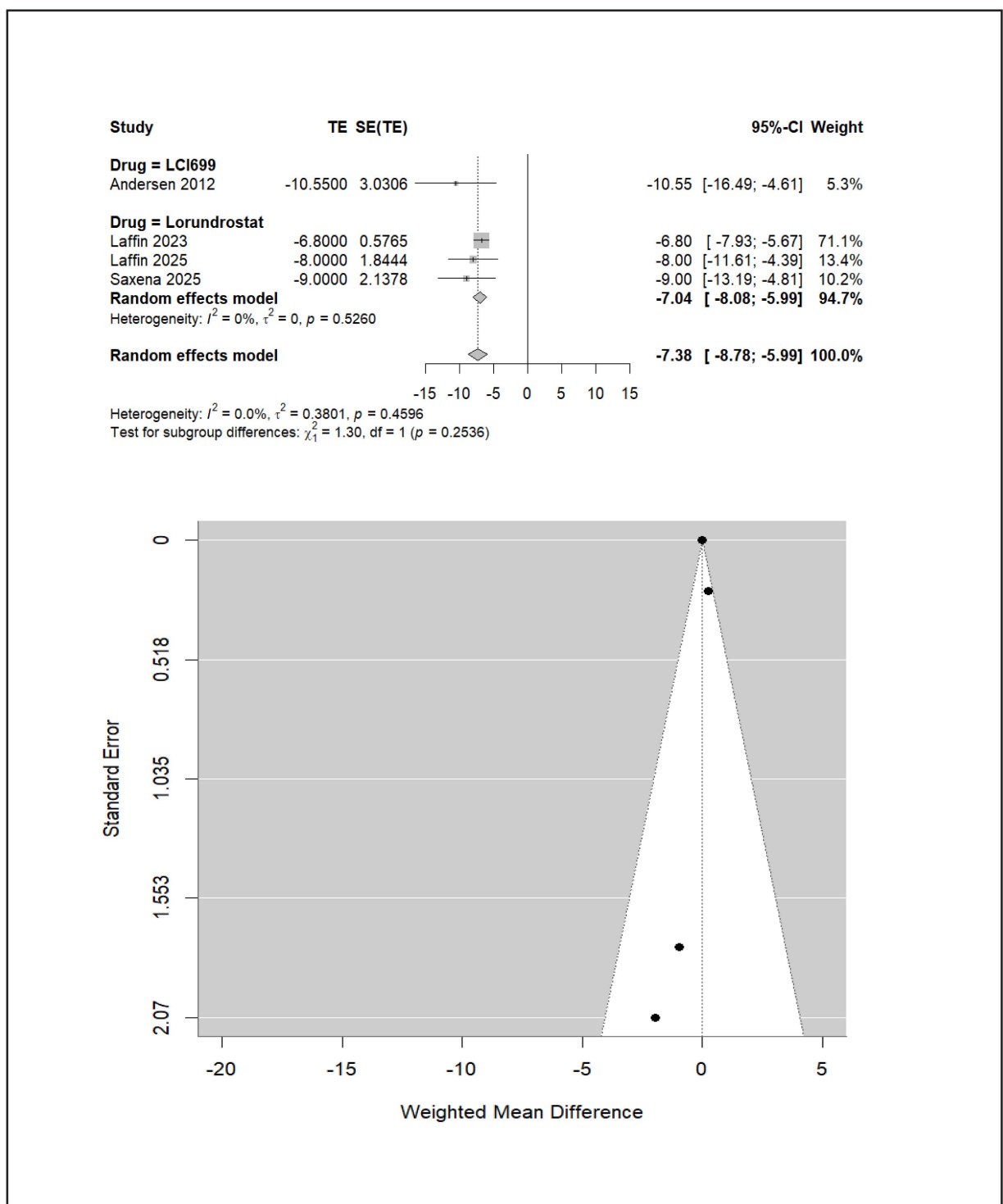

**Fig 6. Uncontrolled hypertension funnel and forest plot.**

**Table 3. Dose-response summary across included aldosterone synthase inhibitor trials.**

| Study | Drug | Population | Doses Studied | Dose with Greatest Placebo-Adjusted Effect | Placebo-Adjusted SBP Change (mmHg) |
|---|---|---|---|---|---|
| Calhoun 2011 | LCI699 | Primary HTN | 0.5 mg, 1 mg QD | 1 mg QD | Not fully reported (DBP: −7.1 vs −2.6 mmHg for placebo) |
| Karns 2013 | LCI699 | Resistant HTN | 0.5 mg QD, 0.5→1 mg BID | 1 mg QD/ 0.5→1 mg BID | ~−4.3 mmHg (NS) |
| BrigHTN 2023 | Baxdrostat | Resistant HTN | 0.5 mg, 1 mg, 2 mg QD | 2 mg QD | −11.0 mmHg |
| Target-HTN 2023 | Lorundrostat | Uncontrolled HTN | 12.5 mg, 50 mg, 100 mg QD; 12.5 mg, 25 mg BID | 50 mg QD | −9.6 mmHg |
| Advance-HTN 2025 | Lorundrostat | Uncontrolled/Resistant HTN | 50 mg fixed vs 50→100 mg titration | 50 mg QD (fixed) | −7.9 mmHg |
| Launch-HTN 2025 | Lorundrostat | Uncontrolled/Resistant HTN | 50 mg fixed vs 50→100 mg escalation | 50 mg QD | −9.1 mmHg |
| BaxHTN 2025 | Baxdrostat | Uncontrolled/Resistant HTN | 1 mg, 2 mg QD | 2 mg QD | −9.8 mmHg |

**Footnotes:**

• Placebo-adjusted SBP changes are derived from reported data in primary trial publications and reflect within-trial dose comparisons, not pooled meta-analytic estimates.

• QD = once daily; BID = twice daily; NS = not statistically significant; HTN = hypertension; SBP = systolic blood pressure; DBP = diastolic blood pressure.

50 mg once daily across all three trials, with escalation to 100 mg providing only marginal additional benefit. For LCI699/osilodrostat, dose-response evidence was limited and non-significant in the resistant hypertension population. These observations are descriptive and hypothesis-generating; formal dose-response characterization will require individual patient data meta-analysis or prospectively designed trials.

### 3.8. Safety profile

Three adverse events were significantly increased compared with placebo. Hypotension was significantly increased (RR 2.67; 95% CI 1.45–4.92; $p = 0.0112$; I² = 0%), with leave-one-out sensitivity analyses yielding stable estimates ranging from RR 2.33 to 2.86, though omitting Laffin 2025 rendered the result marginally non-significant (RR 2.66; 95% CI 0.93–7.56; $p = 0.0587$). Hyperkalemia was significantly increased (RR 7.94; 95% CI 3.03–20.82; $p = 0.0027$; I² = 0%). Individual study estimates were directionally consistent, and leave-one-out analyses remained significant across all exclusions, confirming robustness. This finding is clinically important given the mineralocorticoid pathway targeting of these agents and their anticipated use alongside renin–angiotensin system blockade. Available follow-up was limited to 6–12 weeks; longer-term hyperkalemia risk remains unknown.

Hyponatremia was nominally significantly increased (RR 2.07; 95% CI 1.25–3.41; $p = 0.0138$; I² = 0%), but was heavily dependent on Saxena 2025 (58.8% weight); omitting this trial rendered the result non-significant (RR 1.65; 95% CI 0.70–3.86; $p = 0.1782$). This signal should therefore be interpreted cautiously. Discontinuation due to adverse events was nominally significant (RR 1.87; 95% CI 1.20–2.92; $p = 0.0137$; I² = 0%) but similarly fragile, driven predominantly by Flack 2025 (52.1% weight); omitting it rendered the result non-significant (RR 1.85; 95% CI 0.88–3.88; $p = 0.0848$). Serious adverse events were not significantly increased (I² = 0%). Heterogeneity was negligible across all safety endpoints. Full forest plots, funnel plots, and leave-one-out analyses are provided in the supplementary materials (S4 File).

## 4. Discussion

In this network and pairwise meta-analysis of 7 randomized controlled trials, we evaluated the comparative efficacy and safety of baxdrostat, lorundrostat, and osilodrostat/LCI699 across resistant and uncontrolled hypertension phenotypes.

Several key findings emerged. First, all three agents produced statistically significant and clinically meaningful SBP reductions versus placebo. Second, baxdrostat and lorundrostat demonstrated comparable and pronounced efficacy in resistant hypertension, while osilodrostat/LCI699 showed the largest point estimate in uncontrolled hypertension, though derived from a single trial. Third, DBP reductions mirrored systolic trends with no meaningful between-agent differences. Fourth, short-term safety revealed significantly increased risks of hypotension, hyperkalemia, and hyponatremia, while serious adverse events were not increased. Formal inconsistency testing, heterogeneity assessment, and sensitivity analyses supported the robustness and internal validity of these findings.

### 4.1. Magnitude and clinical relevance of blood pressure reduction

Across the overall network, pooled SBP reductions were −8.63 mmHg for baxdrostat, −7.47 mmHg for lorundrostat, and −5.63 mmHg for osilodrostat/LCI699. Reductions in the 5–10 mmHg range have consistently been associated with substantial decreases in stroke, heart failure hospitalization, and major cardiovascular events [27–33], placing the observed effects, particularly with baxdrostat and lorundrostat, within a clinically consequential range. Indirect comparisons between active agents did not demonstrate statistically significant superiority of any single compound, with all head-to-head confidence intervals crossing zero, reflecting limited power for indirect contrasts rather than equivalence. Ranking analyses under both random-effects and common-effect assumptions were nonetheless consistent: baxdrostat and lorundrostat ranked highest in resistant hypertension, while osilodrostat/LCI699 demonstrated the numerically greatest reduction and highest P-score in uncontrolled hypertension. Given overlapping confidence intervals, ranking probabilities should be interpreted cautiously, though the consistency of directional effects suggests potential phenotype-dependent differences in response. DBP reductions were directionally concordant, with baxdrostat and lorundrostat both significantly reducing DBP, and no meaningful between-agent difference was detected. The parallel systolic and diastolic reductions reinforce the hemodynamic coherence of aldosterone synthase inhibition, suggesting true modulation of volume and vascular tone rather than isolated systolic effects [9,12–17].

### 4.2. Phenotype-specific effects: Resistant versus uncontrolled hypertension

A distinguishing feature of this analysis is the prespecified phenotype-stratified framework. Although resistant and uncontrolled hypertension are often grouped, they represent overlapping but biologically distinct entities [34]. Resistant hypertension is characterized by persistent elevation despite optimized multidrug therapy, frequently driven by sodium retention, neurohormonal activation, and excess aldosterone activity [1,34], whereas uncontrolled hypertension encompasses a broader, mechanistically heterogeneous population including suboptimal adherence, therapeutic inertia, and variable RAAS activation [35–37].

Within resistant hypertension, all three agents achieved statistically significant SBP reductions. Baxdrostat and lorundrostat produced pronounced reductions approaching 9 mmHg, mechanistically consistent with aldosterone-driven pathophysiology. osilodrostat/LCI699 also achieved significance, though of smaller magnitude (−3.53 mmHg; 95% CI −6.95 to −0.11), ranking substantially lower in probability analyses. The significant subgroup difference test ($\chi^2 = 7.52$, df = 2, $p = 0.0233$) confirms that the effect magnitude differs meaningfully across agents in this phenotype. In uncontrolled hypertension, osilodrostat/LCI699 showed the largest point estimate (−10.55 mmHg), while lorundrostat produced consistent and precise reductions across three trials (−7.04 mmHg; 95% CI −8.08 to −5.99). However, the osilodrostat/LCI699 estimate derives from a single early-phase trial (Calhoun 2011), contributing only 5.3% of total weight; the subgroup difference test was non-significant ($\chi^2 = 1.30$, $p = 0.2536$), and baxdrostat contributed no data to this subgroup. The apparent superiority of osilodrostat/LCI699 in uncontrolled hypertension is, therefore, hypothesis-generating and requires prospective validation.

More broadly, these findings underscore the importance of phenotype-guided therapeutic strategies. As options expand, aligning pharmacologic intervention with underlying biology may enhance precision and cost-effectiveness, provided subtype-specific effects are validated in future outcome-driven trials [1].

### 4.3. Mechanistic considerations and safety profile

Aldosterone synthase inhibitors act upstream in the RAAS by selectively inhibiting CYP11B2, thereby reducing aldosterone biosynthesis. This contrasts with mineralocorticoid receptor antagonists (MRAs), which block receptor signaling without suppressing aldosterone production. Upstream inhibition may theoretically attenuate both receptor-mediated and non-receptor-mediated effects, including vascular inflammation, myocardial remodeling, and endothelial dysfunction [27,28,38].

A clinically significant finding is the substantial increase in hyperkalemia (RR 7.94; 95% CI 3.03–20.82), which remained robust across all leave-one-out sensitivity analyses. Although hyperkalemia is a recognized adverse effect of mineralocorticoid receptor antagonists (MRAs) [38–40], the magnitude of the observed risk warrants careful consideration. This effect likely reflects more complete suppression of aldosterone-mediated potassium excretion compared with receptor-level blockade, further compounded by the frequent concomitant use of background renin–angiotensin–aldosterone system (RAAS) inhibitors in the studied populations. As follow-up was limited to 6–12 weeks, the long-term risk of hyperkalemia remains uncertain. Accordingly, routine electrolyte monitoring is essential, particularly in patients with chronic kidney disease, diabetes mellitus, or concurrent RAAS blockade. Hyponatremia was nominally increased (RR 2.07; 95% CI 1.25–3.41); however, this association was largely driven by a single trial and should therefore be interpreted with caution. Hypotension occurred more frequently with active therapy, consistent with effective blood pressure reduction, underscoring the importance of careful dose titration, particularly in patients receiving multiple background antihypertensive agents. Treatment discontinuation was nominally higher but statistically fragile, and there was no significant increase in serious adverse events. While ASIs achieve meaningful blood pressure reduction, their short-term use is associated with clinically relevant electrolyte disturbances requiring proactive monitoring and longer-term characterization in outcome trials.

### 4.4. Positioning within contemporary hypertension management

Current guidelines designate MRAs as preferred fourth-line therapy in resistant hypertension [41]. However, their use is often limited by hyperkalemia risk, endocrine adverse effects, and incomplete suppression of aldosterone synthesis due to feedback mechanisms. Notably, the present analysis demonstrates that ASIs also carry a significant burden of hyperkalemia, underscoring the need for direct comparative trials rather than assumptions of electrolyte superiority.

The clinical role of ASIs remains to be fully defined. They may serve as alternatives in patients intolerant to MRAs, complementary agents in high-risk individuals, or potential replacements if long-term outcome data demonstrate superiority or improved safety [42]. Completion of contemporary phase 3 programs such as BaxHTN and Launch-HTN marks a transition from proof-of-concept to clinical integration. Nevertheless, definitive positioning within treatment algorithms will depend on demonstration of sustained cardiovascular and renal benefit beyond blood pressure reduction alone.

Dose-response patterns across agents have direct implications for future trial design, particularly for head-to-head comparisons with MRAs. Descriptive synthesis of within-trial data reveals consistent attenuation at upper dose ranges across all three agents: baxdrostat plateaus at 2 mg once daily, lorundrostat achieves optimal effect at 50 mg once daily with no population-level benefit from escalation to 100 mg across all three trials, and LCI699/osilodrostat evidence remains limited to modest DBP reductions in primary hypertension. These patterns suggest that future comparative trials should anchor ASI dosing at established optimal doses, rather than maximum tolerated doses, to avoid confounding efficacy signals with dose-dependent adverse effects, particularly hyperkalemia [43,42].

### 4.5. Strengths, limitations, and future directions

This study has several strengths. The combined use of network and pairwise meta-analytic approaches enabled triangulation of inference. Formal design-by-treatment interaction modeling detected no global inconsistency, supporting coherence

of direct and indirect evidence. Prespecified phenotype stratification addressed a clinically meaningful effect modifier. Safety outcomes were synthesized using conservative Hartung–Knapp variance estimation, enhancing reliability in the context of modest study numbers.

Limitations should be acknowledged. The number of available randomized trials remains limited, restricting the precision of indirect comparisons. Follow-up durations were short, precluding assessment of long-term cardiovascular outcomes. Safety definitions were not uniformly standardized across trials, and this may affect the precision and comparability of pooled safety estimates. Although key effect modifiers were evaluated to support the transitivity assumption, unmeasured or incompletely reported covariates, including aldosterone-to-renin ratio, dietary sodium intake, and medication adherence, could not be fully accounted for and represent a residual limitation of indirect comparisons within the network. Additionally, the predominantly star-shaped network limited loop-specific inconsistency testing. Variability in background therapy may have contributed to residual heterogeneity. Ranking probabilities should be interpreted cautiously when effect sizes are similar. The short-term hyperkalemia signal observed warrants evaluation in longer-term trials to characterize its trajectory and clinical significance.

Future research should prioritize large, adequately powered clinical outcome trials to define the long-term efficacy and safety profile of aldosterone synthase inhibitors. Direct comparative studies both among individual agents and against MRAs are warranted to clarify relative effectiveness and tolerability. Biomarker-guided approaches, including stratification by aldosterone–renin ratio, may further refine patient selection and therapeutic response. In addition, dedicated evaluation in high-risk populations—particularly those with chronic kidney disease, diabetes mellitus, and heart failure—will be essential to define broader clinical applicability.

### 4.6. Conclusions

Selective aldosterone synthase inhibition with baxdrostat, lorundrostat, and osilodrostat/LCI699 produces clinically meaningful SBP reductions across resistant and uncontrolled hypertension phenotypes. Baxdrostat and lorundrostat demonstrate comparable efficacy in resistant hypertension, where all three agents achieved statistical significance. Osilodrostat/LCI699 showed the largest point estimate in uncontrolled hypertension, though this derives from a single early-phase trial and requires prospective validation. Short-term use is associated with significantly increased risks of hypotension, hyperkalemia, and hyponatremia — the hyperkalemia signal being particularly robust across sensitivity analyses — necessitating routine electrolyte and hemodynamic monitoring in clinical practice. Serious adverse events were not increased, and discontinuation was nominally elevated but statistically fragile. While aldosterone synthase inhibition represents a mechanistically targeted and promising strategy in difficult-to-treat hypertension, its benefit-risk profile requires careful characterization in longer-term outcome trials. Direct comparative studies against MRAs and adequately powered phenotype-stratified trials will ultimately define its place in contemporary management.

### Supporting information

**S1 File. PROSPERO registration document.**
(PDF)

**S2 File. Search strategy.**
(DOCX)

**S3 File. League tables for all network meta-analyses.**
(PDF)

**S4 File. All safety outcomes: Forest plots, funnel plots, and sensitivity analyses.**
(PDF)

**S5 File. PRISMA checklist.**
(DOCX)

## Author contributions

**Conceptualization:** Ismaila Ajayi Yusuf, Micah Okwah.

**Data curation:** Ismaila Ajayi Yusuf, Alozie Ihesiulo.

**Formal analysis:** Ismaila Ajayi Yusuf.

**Investigation:** Ismaila Ajayi Yusuf, Alozie Ihesiulo, Micah Okwah.

**Methodology:** Ismaila Ajayi Yusuf, Alozie Ihesiulo, Micah Okwah.

**Supervision:** Olurotimi J. Badero.

**Validation:** Ismaila Ajayi Yusuf, Alozie Ihesiulo, Micah Okwah.

**Visualization:** Ismaila Ajayi Yusuf.

**Writing – original draft:** Ismaila Ajayi Yusuf.

**Writing – review & editing:** Ismaila Ajayi Yusuf, Olurotimi J. Badero, Alozie Ihesiulo, Micah Okwah, Abieyuwa Oshodin, Emmanuella Asikong.

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
