## [Decision Letter · Decision Letter 0]

13 Apr 2026

PONE-D-26-09947Aldosterone synthase inhibitors in uncontrolled and resistant hypertension: A phenotype-stratified systematic review and network meta-analysis of randomized trialsPLOS One

Dear Dr. Okwah,

Thank you for submitting your manuscript to PLOS ONE. After careful consideration, we feel that it has merit but does not fully meet PLOS ONE’s publication criteria as it currently stands. Therefore, we invite you to submit a revised version of the manuscript that addresses the points raised during the review process. Please respond to my comments and each of the points made by Reviewer #1.

We look forward to receiving your revised manuscript.

Kind regards,

James M Wright

Academic Editor

PLOS One

Journal Requirements:

Additional Editor Comments:

The major missing information in this systematic review is whether there is a dose response effect for any of these drugs. This will be essential for designing future head to head trials comparing these new agents with mineralacoid receptor antagonists. The different doses studied and included needs to be provided for each of the trials and this issue needs to be discussed in the discussion.

Reviewers' comments:

Reviewer's Responses to Questions

**Comments to the Author**

1. Is the manuscript technically sound, and do the data support the conclusions?

Reviewer #1: Yes

2. Has the statistical analysis been performed appropriately and rigorously? 

Reviewer #1: Yes

3. Have the authors made all data underlying the findings in their manuscript fully available?

Reviewer #1: Yes

4. Is the manuscript presented in an intelligible fashion and written in standard English?

Reviewer #1: Yes

5. Review Comments to the Author

Reviewer #1: This manuscript addresses a clinically important question using a biologically plausible, phenotype‑stratified framework and draws appropriately cautious conclusions. The statistical methods are well chosen for a sparse, largely star‑shaped network and are applied rigorously, including conservative variance estimation, appropriate effect measures for safety outcomes, and a clear distinction between heterogeneity and inconsistency. Key limitations are transparently acknowledged, and subgroup inferences are appropriately based on interaction testing rather than naïve comparisons of p‑values.

Several minor clarifications would further strengthen the manuscript.

- In the Abstract, the statement that osilodrostat/LCI699 shows the largest point estimate in uncontrolled hypertension relies on a single early‑phase trial; while this limitation is acknowledged later in the manuscript, additional clarification in the Abstract would help avoid over‑interpretation based on the abstract alone.

- The rationale for pooling multiple dose arms could be more explicitly stated. Although this approach is standard and reasonable in network meta‑analysis when no single target dose is prespecified, a brief explanation of why pooling was appropriate in this context, and whether dose‑specific differences in efficacy or adverse events were considered or appeared minimal, would improve transparency without requiring additional analyses.

- While transitivity is appropriately discussed and baseline comparisons are provided, the authors are encouraged to briefly describe how potential treatment‑effect modifiers (e.g., baseline SBP, hypertension phenotype, background therapy, trial duration) were identified and whether this selection was prespecified based on biological plausibility or prior evidence. A short acknowledgment of possible unmeasured or unreported effect modifiers and their potential impact on indirect comparisons would further strengthen the assessment of transitivity.

-Inclusion of a trial‑level baseline characteristics table, rather than only treatment‑level summaries in Table 1, would further support assessment of transitivity.

-The Results describe an exploratory meta‑regression to investigate sources of heterogeneity; it would be helpful to clarify whether the exploratory meta‑regression was prespecified or conducted post hoc, as it is not introduced in the Methods section.

-Figure 4 may require clarification, as it appears to include LCI699 trials despite the text stating that no extractable DBP data were available for these studies; outcome‑specific network figures should include only treatments contributing data for that outcome, or the figure and caption should be revised accordingly.

-Including supporting references for the thresholds for lines  293–295 would be helpful.

-Clarification regarding adverse‑event grading would be helpful, including whether events were captured irrespective of severity, whether standardized or trial‑specific grading systems were used, and how heterogeneity in AE grading across trials was handled.

-Minor formatting inconsistencies in confidence intervals and use of dashes; suggest to have these corrected.

Overall, these are minor, clarificatory points that do not require additional analyses and do not detract from the overall strength of the manuscript.

6. PLOS authors have the option to publish the peer review history of their article (what does this mean?). If published, this will include your full peer review and any attached files.

Reviewer #1: No

---

## [Author Response · Author response to Decision Letter 1]

4 May 2026

We thank the Academic Editor and Reviewer #1 for their careful and constructive evaluation of our manuscript. We have addressed all comments in a detailed, point-by-point response provided in the uploaded “Response to Reviewers” document. All corresponding revisions have been incorporated into the manuscript and are clearly indicated in the tracked-changes version.

---

## [Editor Report · Decision Letter 1]

8 May 2026

Aldosterone synthase inhibitors in uncontrolled and resistant hypertension: A phenotype-stratified systematic review and network meta-analysis of randomized trials

PONE-D-26-09947R1

Dear Dr. Okwah,

We’re pleased to inform you that your manuscript has been judged scientifically suitable for publication and will be formally accepted for publication once it meets all outstanding technical requirements.

Kind regards,

James M Wright

Academic Editor

PLOS One
---

## [Editor Report · Acceptance letter]

PONE-D-26-09947R1

PLOS One

Dear Dr. Okwah,

I'm pleased to inform you that your manuscript has been deemed suitable for publication in PLOS One. Congratulations! Your manuscript is now being handed over to our production team.

Kind regards,

on behalf of

Professor James M Wright

Academic Editor

PLOS One